# RETHINKING ADVERSARIAL TRAINING WITH NEURAL TANGENT KERNEL

## ABSTRACT

Adversarial training (AT) is an important and attractive topic in deep learning security, exhibiting mysteries and odd properties. Recent studies of neural network training dynamics based on Neural Tangent Kernel (NTK) make it possible to reacquaint AT and deeply analyze its properties. In this paper, we perform an in-depth investigation of AT process and properties with NTK, including error-bound analysis and NTK evolution. We uncover three new findings that are missed in previous works. First, we disclose the impact of data normalization on AT and the importance of unbiased estimators in batch normalization layers. Second, we experimentally explore the kernel dynamics and propose more time-saving AT methods. Third, we study the spectrum feature inside the kernel to address the catastrophic overfitting problem. To the best of our knowledge, it is the first work leveraging the observations of kernel dynamics to improve existing AT methods.

## 1 INTRODUCTION

In the past few years, Neural Tangent Kernel (NTK) has become a prevalent tool to analyze and understand deep learning models (i.e., neural networks). To study the optimization process in the function space, Jacot et al. (2018) proved that the dynamics of an infinite-wide neural network under gradient descent can be computed with NTK. Lee et al. (2019) further found that the learning process of such a neural network can be approximated as a linear model with the initial values of its parameters. Atanasov et al. (2022) proved that neural networks can be regarded as kernel machines, whose learning dynamics can be divided into two phases: in the "kernel learning" stage, NTK of a neural network is aligned with the features of the network in a tangent space; in the "lazy training" phase, the kernel stays static, and the model parameters do not change significantly. Besides the theoretical studies of NTK properties, there are also empirical investigations on NTK of finite-wide networks trained on practical datasets to explore the generalization ability (Fort et al., 2020; Ortiz-Jiménez et al., 2021) and learning process (Baratin et al., 2021) of networks.

The NTK theory also inspires many interesting applications. For example, Zhou et al. (2021) proposed a new meta-learning paradigm based on NTK to improve the generalization of the model to other domains. Yuan & Wu (2021) adopted the "lazy training" property of neural networks and crafted poisons to decrease the victim model's performance under a black-box threat model. Hayase & Oh (2022) leveraged NTK to decrease the ratio of modified data in backdoor attacks. In this paper, we aim to investigate and enhance adversarial training (AT) and model robustness with NTK.

It is found that neural networks are fragile to a kind of data, dubbed adversarial examples (AEs), which mislead the models to make wrong predictions (Goodfellow et al., 2015; Szegedy et al., 2014). It is essential to explore how to defend against AEs and improve the model's robustness. An important and effective defense approach is adversarial training (AT) (Madry et al., 2018), which generates AEs on-the-fly to train the model. However, AT can cause the model to have lower clean accuracy and overfit AEs on the test set in an earlier training stage (Rice et al., 2020). A lot of efforts have been devoted to the exploration and explanation of such odd property (Dong et al., 2022; Ilyas et al., 2019; Li et al., 2020; Schmidt et al., 2018). Recently, there are two works (Loo et al., 2022; Tsilivis & Kempe, 2022) that leverage NTK to study AT. Loo et al. (2022) empirically studied the dynamics of finite-wide neural networks under normal training and AT. They found that compared with normal training, AT can make the kernel converge to a different one inheriting the model robustness at a much higher speed, and the corresponding eigenvectors of NTK can reflect more visually interpretable features. Tsilivis & Kempe (2022) studied the adversarial robustness by analyzing the features

contained in the neural network's NTK. They empirically proved the transferability of AEs directly generated from NTKs by treating the neural networks as kernel machines (Atanasov et al., 2022). They also studied the spectrum of NTKs and found the robust features (Ilyas et al., 2019) and useful features (Ilyas et al., 2019) can be recognized from the contribution to adversarial perturbation.

However, the above two works exhibit the following weaknesses. (1) They lack a theoretical analysis of NTK for modern neural networks with AT. (2) They only focus on the early training stage but do not consider the characteristics of NTK in the entire AT process. (3) They just analyze one typical AT strategy (Madry et al., 2018), and the applicability to other solutions is unknown. (4) These studies only provide some analysis of AT, but not practical solutions to enhance existing methods.

Motivated by these limitations, this paper presents a more comprehensive analysis of AT with NTK from both theoretical and empirical perspectives. First, we observe there exists an error gap between the ground-truth NTK and empirical NTK in AT. We theoretically analyze the dynamics of neural networks under AT and give an error bound for such a gap (Section 2). Second, we observe that NTK changes significantly in the AT process. So, we explore the full training traces of neural networks with normal training and various AT strategies to reveal the evolution of NTK (Section 3).

Our analysis can deepen our understanding of AT, and more importantly, provide new strategies to enhance existing AT methods. We provide three case studies. (1) Inspired from the error gap analysis, we show how batch normalization (Ioffe & Szegedy, 2015) can affect the clean accuracy and robust accuracy in AT (Section 4). (2) By studying the NTK evolution process, we propose a new training paradigm to reduce the time cost of AT without sacrificing the model robustness and performance (Section 5). (3) We also prove that catastrophic overfitting (Kim et al., 2021) in the single-step AT is sourced from the isotropic features, and provide a simple solution to address this problem (Section 6).

## 2 ANALYSIS OF NTK ERROR GAP IN AT

Prior studies empirically calculate NTK (i.e., ENTK) in AT based on the AEs involved in training (Loo et al., 2022; Tsilivis & Kempe, 2022). However, this ENTK can be different from the ground-truth NTK, as the model parameters could affect the generated AEs. It is important to study the gap between the ground-truth NTK and ENTK in AT to avoid the misinformation from the errors if existed. In this section, we give theoretical analysis of such a gap. Without loss of generality, our study focuses on AT in 2D image classification tasks, which is the same as previous works (Loo et al., 2022; Tsilivis & Kempe, 2022).

### 2.1 NOTATIONS

**Neural Tangent Kernel**. Consider a neural network $f : \mathbb{R}^{n_{\text{in}}} \to \mathbb{R}^{n_{\text{out}}}$ parameterized by a vector of parameters $\theta$. To study the evolution of $\theta$ in the training phase, we rewrite it as $f_t = f(\cdot, \theta_t)$ to represent the specific training state at the time $t$. To describe the changes in parameters and outputs of $f$ on the training set $\mathcal{D} = \{(x, y)|x \in \mathbb{R}^{n_{\text{in}}}, y \in \mathbb{R}^{n_{\text{out}}}\}$, we further define a loss function $\mathcal{L} = \mathbb{E}_{(x,y) \sim \mathcal{D}} \ell(f_t(x), y)$ and a learning rate $\eta$ for parameter update. Then, we have

$$\Delta(\theta)_t = \theta_t - \theta_{t-1} = -\eta \nabla_{\theta_{t-1}} f_{t-1}(\mathcal{X})^T \nabla_{f_{t-1}(\mathcal{X})} \mathcal{L}$$

$$\Delta(f(\mathcal{X}))_t = f_t(\mathcal{X}) - f_{t-1}(\mathcal{X}) = \nabla_{\theta_{t-1}} f_{t-1}(\mathcal{X}) \Delta(\theta)_t,$$

where $\mathcal{X} = \{x|(x, y) \in \mathcal{D}\}$. $f(\mathcal{X})$ represents a matrix with the shape of $|\mathcal{X}|n_{\text{out}} \times 1$, i.e., the outputs of all training data. We find that $\Delta(f(\mathcal{X}))_t = -\eta \nabla_{\theta_{t-1}} f_{t-1}(\mathcal{X}) \nabla_{\theta_{t-1}} f_{t-1}(\mathcal{X})^T \nabla_{f_{t-1}(\mathcal{X})} \mathcal{L}$. This enables us to deduce NTK (Jacot et al., 2018) as: $\Theta(\mathcal{X}, \mathcal{X})_{t-1} = \nabla_{\theta_{t-1}} f_{t-1}(\mathcal{X}) \nabla_{\theta_{t-1}} f_{t-1}(\mathcal{X})^T$, which reflects that how the changes of model parameters with respect to some data affect the predictions of other (or same) data. So, the shape of $\Theta$ is $|\mathcal{X}|n_{\text{out}} \times |\mathcal{X}|n_{\text{out}}$. To be aligned with the formation in deep learning tools (e.g., Pytorch (Paszke et al., 2019), JAX (Bradbury et al., 2018)), we reformat NTK to have the shape of $|\mathcal{X}| \times |\mathcal{X}| \times n_{\text{out}} \times n_{\text{out}}$.

**Adversarial Training**. In AT, AEs are generated and participate in model training. We denote $x \in \mathcal{X}$ as a clean sample, and the corresponding $x' \in \mathcal{B}_p^\epsilon(x)$ as an AE, if it satisfies $\arg \max f(x) \neq \arg \max f(x')$, where $\mathcal{B}_p^\epsilon(x)$ is a $l_p$-norm ball with the center of $x$ and radius of $\epsilon$. Generally, searching such an AE from $x$ is based on the gradients with respect to $x$, i.e., $x' = x + \alpha \nabla_x \ell(f(x), y)$, where $\alpha$ can be seen as a learning rate for $x$. Therefore, the training object of AT is to minimize the empirical loss $\mathbb{E}_{(x,y) \sim \mathcal{D}} \ell(f_t(x'), y)$ at epoch $t$.

## 2.2 ERROR GAP ANALYSIS

Section 2.1 gives the definition of NTK for a model trained with clean data. In this case, input data are independent of model parameters. However, in AT, input data (AEs) are related to the current model parameters obeying the following connection:

$$x' = x + \omega \quad s.t. \quad \omega^{i,j,k} \in \{-\epsilon, \epsilon\}, \ \mathrm{P}(\omega^{i,j,k} = \epsilon) = \mathrm{P}(\nabla_x \ell(f(x), y)^{i,j,k} > 0),$$

where $\omega \in \{-\epsilon, \epsilon\}^3$ is the perturbation added to the clean sample $x$, which is an RGB image. Clearly, the perturbation for each pixel only depends on the gradient with respect to the corresponding pixel. Note that we consider the perturbation is under $l_\infty$-norm. The generated AEs are usually clipped into $[0, 1]$ to fit the model's input range. However, for simplicity, we omit this step and set the perturbation at each pixel as either $-\epsilon$ or $\epsilon$, which is aligned with the practical observation that the perturbation reaches either the maximum or minimum. For a multi-step attack, such as PGD, AEs are generated by overlaying multi-round gradients. In such a case, $\nabla_x \ell(f(x), y)^{i,j,k}$ represents the overlaid gradients.

So, this gives us two aspects to study NTK of a model with AT: clean sample NTK and AE NTK. Based on the definition in Section 2.1, in AT, we have

$$\Delta(\theta)_t = \theta_t - \theta_{t-1} = -\eta \nabla_{\theta_{t-1}} f_{t-1}(\mathcal{X} + \Omega_{t-1})^T \nabla_{f_{t-1}(\mathcal{X} + \Omega_{t-1})} \mathcal{L}_{\mathrm{AT}}$$

$$\Delta(f(\mathcal{X}))_t = f_t(\mathcal{X}) - f_{t-1}(\mathcal{X}) = \nabla_{\theta_{t-1}} f_{t-1}(\mathcal{X}) \Delta(\theta)_t$$

$$\Delta(f(\mathcal{X} + \Omega_{t-1}))_t = f_t(\mathcal{X} + \Omega_{t-1}) - f_{t-1}(\mathcal{X} + \Omega_{t-1}) = \nabla_{\theta_{t-1}} f_{t-1}(\mathcal{X} + \Omega_{t-1}) \Delta(\theta)_t,$$

where $\Omega_{t-1} = [\omega_1, \omega_2, \ldots, \omega_{|\mathcal{X}|}]$ is all perturbation for $x_i \in \mathcal{X}$ at epoch $t-1$, and $\mathcal{L}_{\mathrm{AT}} = \mathbb{E}_{(x,y) \sim \mathcal{D}} \ell(f_t(x + \omega), y)$. Clearly, NTK of $x'$ can be calculated as $\Theta(\mathcal{X} + \Omega_{t-1}, \mathcal{X} + \Omega_{t-1})_{t-1} = \nabla_{\theta_{t-1}} f_{t-1}(\mathcal{X} + \Omega_{t-1}) \nabla_{\theta_{t-1}} f_{t-1}(\mathcal{X} + \Omega_{t-1})^T$. On the other hand, NTK of $x$ can be calculated as $\Theta(\mathcal{X} + \Omega_{t-1}, \mathcal{X})_{t-1} = \nabla_{\theta_{t-1}} f_{t-1}(\mathcal{X} + \Omega_{t-1}) \nabla_{\theta_{t-1}} f_{t-1}(\mathcal{X})^T$. Considering that in normal training, we only have $\Theta(\mathcal{X}, \mathcal{X})_{t-1} = \nabla_{\theta_{t-1}} f_{t-1}(\mathcal{X}) \nabla_{\theta_{t-1}} f_{t-1}(\mathcal{X})^T$. Therefore, we wish to explore the implicit connection between $\Theta(\mathcal{X} + \Omega_{t-1}, \mathcal{X})_{t-1}$ and $\Theta(\mathcal{X}, \mathcal{X})_{t-1}$ in adversarial training.

**Theorem 1** *Given a $\mathcal{C}^2$ function $f : \mathbb{R}^{n_{\mathrm{in}}} \to \mathbb{R}^{n_{\mathrm{out}}}$, parameterized by a vector of parameters $\theta$, and trained by $x' = x + \omega$, where $\omega \in \{-\epsilon, \epsilon\}^3$, on a dataset $\mathcal{D} = \{(x, y) | x \in \mathbb{R}^{n_{\mathrm{in}}}, y \in \mathbb{R}^{n_{\mathrm{out}}}\}$, NTK of $\mathcal{X}$ satisfies $\Theta(\mathcal{X} + \Omega_{t-1}, \mathcal{X})_{t-1} \approx \Theta(\mathcal{X}, \mathcal{X})_{t-1} + o(\epsilon)(\epsilon \to 0)$, where $\mathcal{X} = \{x | (x, y) \in \mathcal{D}\}$ and $\Omega_{t-1} = [\omega_1, \omega_2, \ldots, \omega_{|\mathcal{X}|}]$, which is a random perturbation set independent of $x$, $\theta$, and $\ell$.*

Proof can be found in Appendix A. Because it is very complex if we consider exactly the same setting as the adversarial training, where the perturbation is highly related to the $x$, $\theta$, and $\ell$, we make some strong assumptions in our proof to simplify it. From Theorem 1, we know adversarial training will make NTK of $x$ shift. Furthermore, if $\epsilon$ is small in AT, e.g., $8/255$, the clean accuracy will not drop too much. However, if the perturbation size is large, NTK of $x$ will significantly change, which could explain the reason that using a large perturbation size will hurt the clean accuracy.

In the above theorem, we assume the parameter update in AT is over the entire dataset. However, in practice, the training process is usually based on mini-batches. To study a kernel of models trained with mini-batches, we have the following theorem.

**Theorem 2** *Given a $C$-Lipschitz function $f : \mathbb{R} \to \mathbb{R}$ defined on a $l_2$-normed metric space, suppose $\mathcal{X} = \{x | x \in \mathbb{R}\}$ is the input set, whose cardinality is $|\mathcal{X}|$. Suppose there are $k$ disjoint subsets of $\mathcal{X}$ having the same size, i.e., $\tilde{\mathcal{X}}_i \in \mathcal{X}, i \in [k]$, and $|\tilde{\mathcal{X}}_i| = \alpha \leq |\mathcal{X}|$. Let $S_i^1 = \sum_{j=1}^{\alpha} x_j$ and $S_i^2 = \sum_{j=1}^{\alpha} x_j^2$, for $x_j \in \tilde{\mathcal{X}}_i$, and $A = S_i^2 - (S_i^1)^2$. For $x \in \tilde{\mathcal{X}}_i$, we have*

$$\|f(\frac{x - \tilde{\Sigma}_i}{\tilde{\Pi}_i}) - f(\frac{x - \Sigma}{\Pi})\|_2 \leq C\|x\|_2(\|\frac{1}{\tilde{\Pi}_i}\|_2 + \|\frac{\alpha}{\sqrt{\mathbb{E}(A)}}\|_2) + C(\|\frac{\tilde{\Sigma}_i}{\tilde{\Pi}_i}\|_2 + \|\frac{\mathbb{E}(S_i^1)}{\sqrt{\mathbb{E}(A)}}\|_2),$$

*where $\tilde{\Sigma}_i$ and $\tilde{\Pi}_i$ are mean and standard variance for elements in $\tilde{\mathcal{X}}_i$, and $\Sigma$ and $\Pi$ are unbiased estimators of mean and standard variance for elements in $\mathcal{X}$.*

Proof can be found in Appendix A. From Theorem 2, for a neural network with normalization (e.g., batch normalization (Ioffe & Szegedy, 2015)), the computation error of a kernel $f$ between mini-batch training and full dataset training mainly depends on the gap between the mean and variance of the

batch and dataset. Specifically, in Theorem 2, suppose the neural network is equipped with batch normalization layers, which is very common in modern models, the kernel of a model trained with mini-batch is asymptotically closed to the kernel of a model trained directly on all data, regardless of the training strategies (normal or AT). Inspired by such analysis, we will further study how batch normalization can effect the model robustness in Section 4.

## 3 ANALYSIS OF NTK EVOLUTION IN AT

Prior studies (Loo et al., 2022; Tsilivis & Kempe, 2022) only focus on NTK at the early stage of AT. However, it can change along with the training process. We experimentally demonstrate the evolution of NTK with different AT strategies, which can shed light on the improvement strategies.

### 3.1 EXPERIMENT CONFIGURATIONS

**Setup**. Due to the overwhelming computation complexity, it is too slow to compute the full NTK on the entire dataset. Instead, we randomly sample some data from the dataset to calculate ENTK (Loo et al., 2022). We use NTK $\Theta(\cdot, \cdot)$ to represent ENTK, if no ambiguity in the followings. Specifically, we randomly select 500 data evenly from all classes to compute ENTK. Similar to previous works (Loo et al., 2022; Tsilivis & Kempe, 2022), we calculate NTK after every training epoch, instead of every gradient descent step. For kernels calculated on clean samples and AEs, we call them C-NTKs and AE-NTKs, respectively.

We consider normal training and three specific AT methods under $l_\infty$-norm (FGSM-AT, PGD-AT (Madry et al., 2018), and TE (Dong et al., 2022)). We choose CIFAR-10 and CIFAR-100 datasets, ResNet-18 model (He et al., 2016) and SGD optimizer (Ruder, 2016). The total number of training epochs is 200, and the weight decay is 0.0005. The learning rate at the beginning is 0.1, and is decayed at the 100-th and 150-th epochs by a factor of 0.1. The batch size is 128. We further use the most common data augmentation methods over the training set: random cropping and random horizontal flipping. For FGSM-AT, we adopt FGSM (Goodfellow et al., 2015) to generate AEs, with hyperparameters $\epsilon = 8/255$ and $\alpha = 8/255$. For PGD-AT and TE, we adopt PGD-10 to generate AEs, with hyperparameters $\epsilon = 8/255$, $\alpha = 2/255$, and 10 steps. For robustness evaluation, we choose PGD-20 to generate AEs on the test set under $l_\infty$-norm. All experiments are conducted with the *functorch* library of Pytorch-2.0. Additionally, in the appendix, we study how model initialization methods can influence kernel dynamics to generalize our conclusions to other deep learning frameworks.

**Metrics**. As aforementioned, NTK has the shape of $|\mathcal{X}| \times |\mathcal{X}| \times n_{\text{out}} \times n_{\text{out}}$. In most cases, we mainly care about the kernel for $f^i(\mathcal{X})$, i.e., the specific output for class $i$ in NTK. The corresponding entry in NTK is described by $\Theta(\mathcal{X}, \mathcal{X})_t^{i,i} = \nabla_{\theta_t} f_t^i(\mathcal{X}) \nabla_{\theta_t} f_t^i(\mathcal{X})^T$, which is a square matrix with the shape of $|\mathcal{X}| \times |\mathcal{X}|$. To study all classes in the dataset $\mathcal{D}$, we average the corresponding entries in NTK, i.e., $\hat{\Theta}(\mathcal{X}, \mathcal{X})_t = \frac{\sum_{i=1}^{n_{\text{out}}} \Theta_t^{i,i}}{n_{\text{out}}}$, where $n_{\text{out}}$ is the number of classes in $\mathcal{D}$. $\hat{\Theta}(\mathcal{X}, \mathcal{X})_t$ is called the "*traced kernel*" in previous works (Shan & Bordelon, 2021; Loo et al., 2022). We adopt three kernel metrics proposed by prior works:

1. *Kernel Distance* (Loo et al., 2022): this is defined as the similarity between two kernels, i.e.,

$$\mathbf{KD}(\hat{\Theta}_{t_1}, \hat{\Theta}_{t_2}) = 1 - \frac{\text{Tr}(\hat{\Theta}_{t_1}^T \hat{\Theta}_{t_2})}{\|\hat{\Theta}_{t_1}\|_F \|\hat{\Theta}_{t_2}\|_F}$$

where $\text{Tr}(\cdot)$ is the trace of the square matrix, and $\|\cdot\|_F$ is the Frobenius norm for the matrix.

2. *Kernel Effective Rank* (Roy & Vetterli, 2007): for a kernel's effective rank, we first do singular value decomposition (SVD) on it, i.e., $\hat{\Theta}_t = U\Sigma V^*$, where $U$ and $V$ are two unitary matrices with the shape of $|\mathcal{X}| \times |\mathcal{X}|$, $V^*$ is the conjugate transpose of $V$, and $\Sigma$ is a diagonal matrix containing singular values. Suppose the singular values satisfy $\sigma_1 \geq \sigma_2 \geq \cdots \geq \sigma_n > 0$, then the effective rank is defined as follows:

$$\mathbf{KER}(\hat{\Theta}_t) = \exp(-\sum_{i=1}^{n} p_i \log p_i),$$

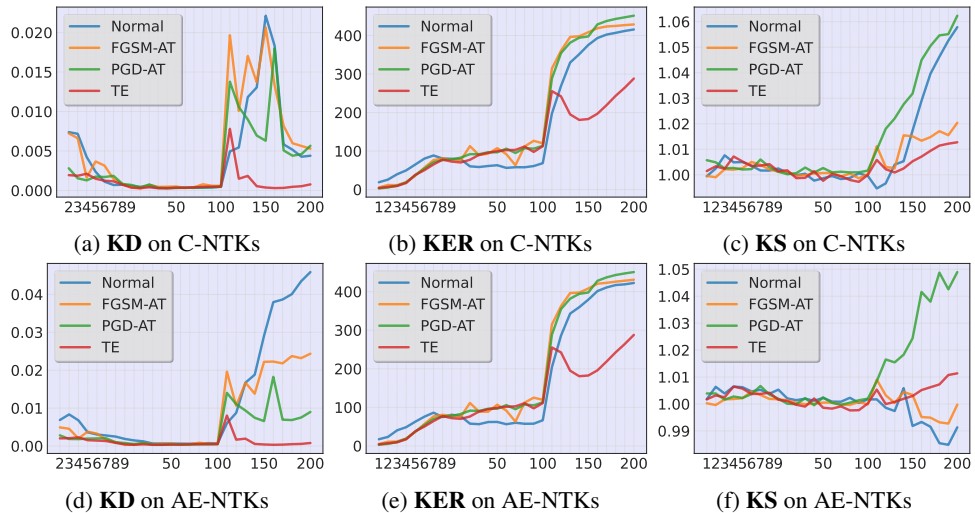

Figure 1: **KD**, **KER**, and **KS** in training. x-axis: training epoch; y-axis: metrics.

where $p_i = \frac{\sigma_i}{\sum_{j=1}^n \sigma_j}$. A large value of $\mathbf{KER}(\hat{\Theta}_t)$ implies that features in the kernel have more equal importance or the dataset is more complex (Loo et al., 2022). Note that as we calculate NTK with respect to $\mathcal{X}$ and $\mathcal{X}$, $\hat{\Theta}_t$ is a positive semi-definite square matrix, which implies that the singular values in $\Sigma$ equal the eigenvalues of $\hat{\Theta}_t$. So, we use eigenvalues to represent the singular values in some cases to align with previous works' conclusions.

3. *Kernel Specialization* (Shan & Bordelon, 2021): this metric studies how the kernel alignment (Cortes et al., 2012) appears and evolves during the training process. We first define a matrix $M_{\mathrm{ks}}(\Theta_t)$ to represent the specialization strengths for a pair of two different classes $(i, j)$ in NTK $\Theta_t$:

$$M_{\mathrm{ks}}^{i,j}(\Theta_t) = \frac{A(\Theta_t^{i,i}, \mathcal{Y}_j \mathcal{Y}_j^T)}{\frac{1}{n_{\mathrm{out}}} \sum_{l=1}^{n_{\mathrm{out}}} A(\Theta_t^{l,l}, \mathcal{Y}_j \mathcal{Y}_j^T)},$$

where $A(\Theta_t^{i,i}, \mathcal{Y}_j \mathcal{Y}_j^T) = 1 - \mathbf{KD}(\Theta_t^{i,i}, \mathcal{Y}_j \mathcal{Y}_j^T)$, and $\mathcal{Y}_j$ is a matrix of shape $|\mathcal{X}| \times 1$. The element in $\mathcal{Y}_j$ is 1 if the corresponding data $x$ has label $j$ or 0 otherwise. So, $M_{\mathrm{ks}}(\Theta_t)$ has the shape of $n_{\mathrm{out}} \times n_{\mathrm{out}}$. Then we deduce the kernel specialization from the corresponding $M_{\mathrm{ks}}(\Theta_t)$:

$$\mathbf{KS}(\Theta_t) = \frac{\mathrm{Tr}(M_{\mathrm{ks}}(\Theta_t))}{n_{\mathrm{out}}}.$$

Shan & Bordelon (2021) proved that the kernel evolves different subkernels for each class $i$ to accelerate the learning process, and each subkernel is aligned with the target function with respect to its class. Therefore, the kernel specialization occurs if the kernel consists of different subkernels. The larger value of $\mathbf{KS}(\Theta_t)$ is, the more severely kernel specialization happens in the kernel, which means that the model is mainly influenced by the feature of the target class.

### 3.2 THREEFOLD EVOLUTION OF NTK IN AT

In Figure 1, we show the training dynamics on CIFAR-10 using three metrics: **KD** between training epochs $t$ and $t-1$, **KER** at training epoch $t$, and **KS** under clean label (CL), which are the ground-truth labels for the input data, at training epoch $t$. We show how these metrics help us better understand AT and normal training. For **KD**, it is clear that before the learning decay at the 100-th epoch, the kernel distance is close to zero for all training strategies after a very quick "kernel learning" process (Figures 1a and 1d). The "lazy learning" appears between the 10-th epoch and 100-th epoch, where the kernel stays almost unchanged. After the learning rate decay, the kernel sharply changes until the training completes at a smaller learning rate, which can be seen as an additional kernel learning phase (Shan & Bordelon, 2021). On the other hand, AE-NTKs show significant differences between these training approaches. But AE-NTKs of models with AT have smaller **KD** than the ones with normal training, which indicates the kernel converges to a specific robust one (Loo et al., 2022).

This "kernel learning"-"lazy training"-"kernel learning" process can be found from **KER** in Figures 1b and 1e as well. In detail, **KER** quickly increases at a very early training stage (before the first 10

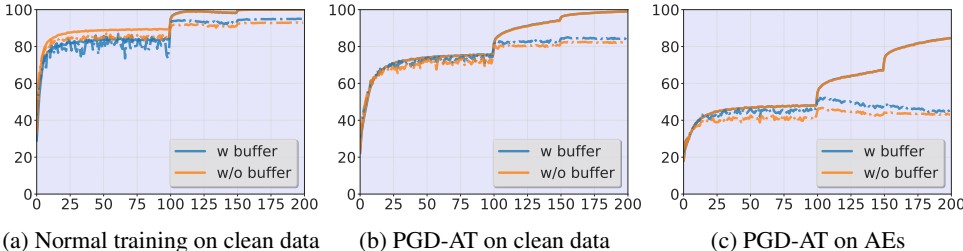

(a) Normal training on clean data    (b) PGD-AT on clean data    (c) PGD-AT on AEs

Figure 2: Accuracy in different settings. x-axis: training epoch; y-axis: accuracy. Solid and dash lines are for training set and test set accuracy, respectively.

epochs), then keeps stable (between the 10-th epoch and 100-the epoch), and finally increases after the learning rate decay. Furthermore, we conclude that **KER** cannot reflect the robustness or other robust-related properties of the models. This is different from the previous work (Loo et al., 2022), which only studied the first 100 training epochs to obtain the conclusion that the robust model's kernel has higher **KER**. Specifically, before the first 100 epochs, **KER** of the robust model's kernels is higher. After the learning rate decay, the kernels of models with PGD-AT, FGSM-AT, and normal training have similar **KER**, which is higher than the value of the model with TE. Therefore, we argue that **KER** cannot really or correctly reflect the robustness of the model.

For **KS**, we find that it can reflect the model's robustness on AE-NTKs. Specifically, if **KS** on AE-NTKs is close to 1, the corresponding models will have higher robustness. However, **KS** on C-NTKs does not have this property. We hypothesize this is because robust models consider features from other classes to classify AEs, while non-robust models mainly depend on class-related features. On the other hand, it is surprising that **KS** on C-NTKs and AE-NTKs disagree, indicating that the models adopt different features (Ilyas et al., 2019; Tsilivis & Kempe, 2022) to predict the clean samples and AEs. This will be an interesting future direction to improve model's clean accuracy under AT. One possible explanation is that the feature selection phenomenon (Baratin et al., 2021) is individually applied to clean samples and AEs.

From the above observations, we find that the kernels of models trained with different methods are frozen at the early training stage ("lazy training"). We further provide a case study on TRADES (Zhang et al., 2019) in Appendix D, and obtain the same conclusion. This provides an opportunity to reduce the training cost of different AT methods. We provide a case study in Section 5 to prove the possibility.

## 4   CASE STUDY I: IMPACT OF BATCH NORMALIZATION ON AT

In Section 2, we analyze the error gap between ENTK and ground-truth NTK. Especially, in Theorem 2, when $f$ represents a neural network equipped with batch normalization layers, since we can approximately estimate $\Sigma$ and $\Pi$ from all seen training data inside the batch normalization layer, we can further deduce the following propositions:

**Proposition 1** *For a model $f$ with batch normalization layers, if $f$ is evaluated on clean data, $\Sigma$ and $\Pi$ will be close to $\tilde{\Sigma}_i$ and $\tilde{\Pi}_i$, the performance of $f$ will not be influenced if we replace $\Sigma$ and $\Pi$ in the batch normalization layers with $\tilde{\Sigma}_i$ and $\tilde{\Pi}_i$.*

**Proposition 2** *For a model $f$ with batch normalization layers, if $f$ is evaluated on adversarial examples, $\Sigma$ and $\Pi$ will be different from $\tilde{\Sigma}_i$ and $\tilde{\Pi}_i$, because the batch internal statistics information varies with the inputs. The performance of $f$ will be significantly influenced if we replace $\Sigma$ and $\Pi$ in the batch normalization layers with $\tilde{\Sigma}_i$ and $\tilde{\Pi}_i$.*

These two propositions indicate that the unbiased estimators (Ioffe & Szegedy, 2015) of mean and standard variance in batch normalization layers can help the model achieve higher robustness instead of clean accuracy. We empirically prove this argument. Specifically, we analyze the practical effects of batch normalization layers with ResNet-18 on CIFAR-10. To simulate the conditions in Propositions 1 and 2, we remove the unbiased estimators in the batch normalization layers and force them to use the estimations from the current batch, which is called the "w/o buffer". On the other hand, the original behaviors of batch normalization layers is called the "w buffer". We consider two training strategies, i.e., normal training and PGD-AT. Figure 2 illustrates the accuracy of clean samples and AEs for the training set and the test set. We observe that the clean accuracy in the "w buffer" and "w/o buffer" is close to each other on both training and test sets. However, the robust

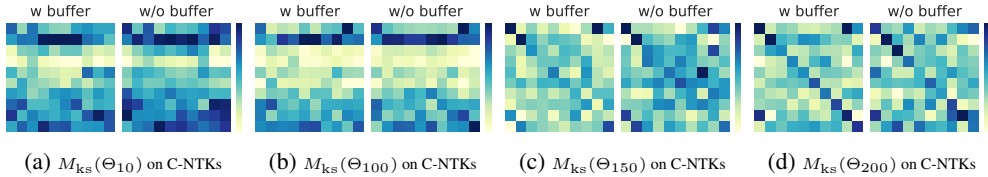

Figure 3: **KS** strengths during PGD-AT under different BN strategies on C-NTKs. Both the x-axis and y-axis represent the classes in CIFAR-10 in the same order, from left to right and from up to down, respectively, which is the same for the other **KS** strength plots.

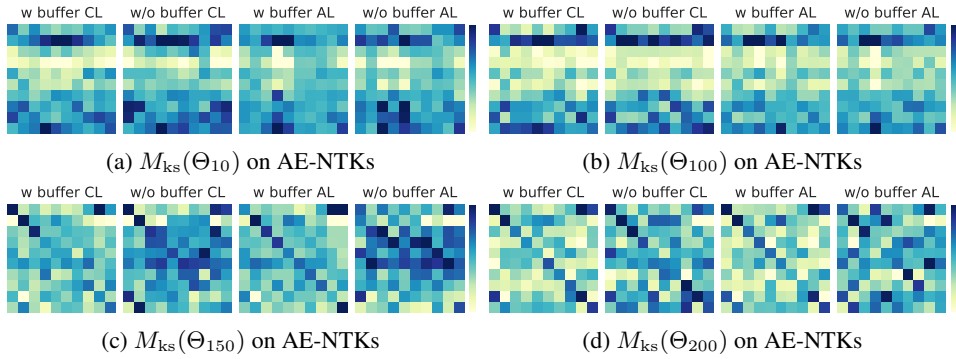

Figure 4: **KS** strengths during PGD-AT under different BN strategies on AE-NTKs.

accuracy on the test set is quite different in the two modes. These results support the aforementioned propositions.

We provide more analysis in the following. Theorem 2 implies that there are two distinct NTKs in the "w buffer" and "w/o buffer", due to the different ways of calculating the statistics. Therefore, we study the NTKs in the two models to explain the underlying reasons for the impacts of batch normalization on the robustness from the feature spectrum in the kernel. Figures 3 and 4 display the kernel specialization process of the model with PGD-AT on clean data and AEs, respectively. The kernel specialization on C-NTKs indicates that the kernel alignment happens in both modes and produce the subkernels with respect to the corresponding class, i.e., having darker color at diagonals in the kernel. So, when we evaluate the model on clean data, both modes can achieve similar results on the training set or the test set. However, when we compare the kernel specialization on AE-NTKs, the alignment effects are significantly different. Figure 4 visualizes the kernel alignment under clean labels (CL), which are the ground-truth labels for the input data, and adversarial labels (AL), which are the model's predicted labels for the input data. First, Figures 4c and 4d indicate that the kernel alignment appears earlier for AL than CL in the "w buffer", which means AEs depend on some specific class-related features and promote the subkernel evolution to decrease the loss under AL. On the other hand, in the "w/o buffer", the kernel alignment is fainter for either CL or AL, which means the model does not heavily depend on the corresponding class's features to predict the input data. It could be because the model gives lower robust accuracy on the test set, as it cannot effectively adopt the most relative features with respect to the correct label to make predictions.

## 5 CASE STUDY II: REDUCING TRAINING COST

In Section 3, we show there is a threefold evolution of NTK during AT. Therefore, it is possible to reduce the training cost by utilizing the "lazy training" phase of the evolution. We provide a detailed case study about this point.

As the training phase is threefold, what we want to know is the kernel differences among these training strategies. To calculate **KD** between two different models, we choose TE as the baseline, i.e., $\text{KD}(\hat{\Theta}_t^{\text{TE}}, \cdot)$, and compare it with other methods. The results in Figure 5 indicate that in the first "kernel learning" stage, different training methods make the kernel converge into the same one, which will be frozen in the "lazy training" stage. However, in the second "kernel learning" stage, the kernels become very different, which means the training strategies mainly influence the last training phase. Therefore, it is natural to investigate whether we can reduce the time cost of AT by using normal training first. The answer is affirmative.

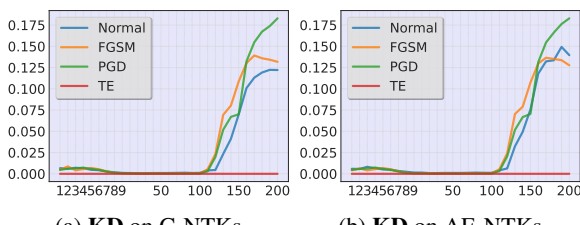

(a) PGD-AT on CIFAR10    (b) TE on CIFAR10    (c) PGD-AT on CIFAR100    (d) TE on CIFAR100

Figure 6: New training paradigm to reduce the time cost of AT. x-axis: training epoch; y-axis: accuracy. Solid and dash lines represent the clean and robust accuracy, respectively.

Figure 6 shows the performance of our improved training strategy on the test set. We consider two AT methods, namely PGD-AT and TE. In these methods, we replace AEs with clean data before the $i$-th epoch. For instance, the notation "PGD-AT 90" indicates that we use only clean data to train the model for the first 90 epochs,

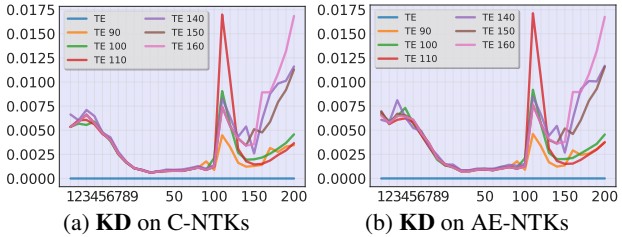

(a) **KD** on C-NTKs    (b) **KD** on AE-NTKs

Figure 5: **KD** between different strategies.

and then adopt PGD-AT till epoch 200. The results indicate that this strategy can reduce the training cost significantly without sacrificing model performance. Specifically, the training time cost for one epoch with AEs is 90 seconds on one RTX 3090, and the time for clean data is 25 seconds. So, the total training cost for PGD-AT and TE is 18,000 seconds. For "PGD-AT 100" and "TE 100", the total time is **11,500** seconds, which is **63.88%** of the original cost. For "PGD-AT 140" and "TE 140", the total time is **8,900** seconds, which is only **49.44%** of the original one. We discuss how to decide a proper time to switch the training mode in Appendix C and provide robust accuracy under various attacks.

Furthermore, we explain why the model can achieve similar performance when we replace AEs with clean data at the early training stage by comparing the kernels of corresponding models. Figure 7 shows **KD** between TE and its variants on CIFAR-10. The values of **KD** are aligned with the model's clean accuracy and robust accuracy. In detail, if **KD** is smaller (e.g., "TE 90", "TE 100", "TE 110"), both clean accuracy

(a) **KD** on C-NTKs    (b) **KD** on AE-NTKs

Figure 7: Kernel differences under different paradigms measured by **KD**.

and robust accuracy of the model are close to the model with TE. If **KD** is larger (e.g., "TE 160"), the model's performance will have a slight decrease.

# 6 CASE STUDY III: ANALYZING AND OVERCOMING CATASTROPHIC OVERFITTING

One special feature of models trained with the one-step adversarial attack is catastrophic overfitting: the models will suddenly lose robustness on the test set. In this section, we explain this phenomenon by comparing the training dynamics of FGSM-AT and TE using NTK, and further propose a simple solution to address it.

First, Figure 8 plots the **KS** strength on AE-NTKs for models trained with different methods on CIFAR-10. It is clearly observed that the kernels of robust models, which are trained with PGD-AT and TE, are totally different from the kernels of non-robust models, which are trained with normal training and FGSM-AT. Specifically, the strength matrices of non-robust models show more chaos under CL. On the contrary, the strength matrices of robust models are aligned along the diagonal and with a specific pattern. These results indicate that catastrophic overfitting occurs because the model only depends on some non-robust features to predict the labels of AEs. However, the root cause that the model trained with FGSM-AT learns such non-robust features is unclear. Here, we make an assumption that the one-step adversarial attack will make some shortcut robust features in the generated AEs, which cannot generalize to the test set.

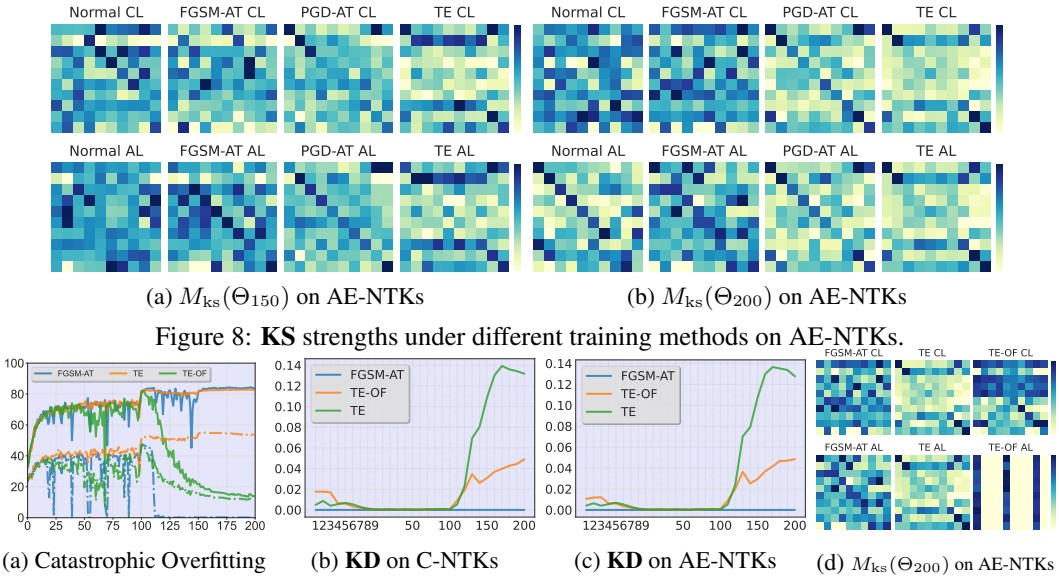

(a) $M_{\mathrm{ks}}(\Theta_{150})$ on AE-NTKs          (b) $M_{\mathrm{ks}}(\Theta_{200})$ on AE-NTKs

Figure 8: **KS** strengths under different training methods on AE-NTKs.

(a) Catastrophic Overfitting    (b) **KD** on C-NTKs    (c) **KD** on AE-NTKs    (d) $M_{\mathrm{ks}}(\Theta_{200})$ on AE-NTKs

Figure 9: Studies of shortcuts in FGSM-AT. In Figure 9a, solid lines represent clean accuracy, and dash lines represent robust accuracy on the test set.

Then, to empirically verify our assumption, we generate such a shortcut by modifying the data augmentation process. By default, all data inside the mini-batch are augmented individually, which means that if we use rotation for augmentation, data inside the mini-batch will be rotated by different angles. In our modified process, all data in the mini-batch will follow the same augmentation setting, e.g., rotation by the same angle. Therefore, the model can find a shortcut from the augmented data that are forced to be aligned in a similar distribution, i.e., isotropic features. We adopt the modified augmentation strategy to train a model with TE (dubbed TE-OF). Figure 9a shows the clean accuracy and robust accuracy on the test set of CIFAR-10. Clearly, the model trained with TE-OF shows a similar overfitting behavior as the model trained with FGSM-AT. We further compare the kernels of different models in Figures 9b and 9c, where we choose the kernels of the model trained with FGSM-AT as the baselines. The values of **KD** indicate that the model trained with TE-OF has a similar kernel as the FGSM-AT model, which is aligned with the observation from Figure 9a. The **KS** strength matrices on AE-NTKs can be found in Figure 9d, from which we find the matrix of TE-OF model is chaotic and looks like the one of the FGSM-AT model.

A straightforward and effective strategy to eliminate such shortcuts is to add some anisotropy noise into every sample inside each mini-batch. In Figure 10, we prove the effectiveness of this strategy with experiments on CIFAR-10 and CIFAR-100. Compared to the models trained with FGSM-AT, the models trained with our proposed variant overcome the catastrophic overfitting problem and achieve higher robustness.

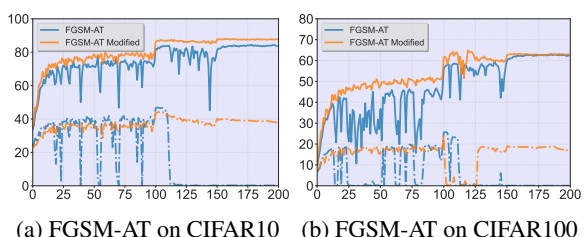

(a) FGSM-AT on CIFAR10    (b) FGSM-AT on CIFAR100

Figure 10: Our solution overcomes catastrophic overfitting. Solid and dash lines represent clean and robust accuracy.

## 7    DISCUSSION AND CONCLUSION

This paper presents a comprehensive analysis of AT with NTK. We disclose the existence of error gap between the ground-truth and empirical NTK in AT, as well as the NTK evolution during training. Based on the analysis, we provide three case studies, which help us better understand the impact of batch normalization on model robustness, reduce the AT cost, and overcome catastrophic overfitting.

On the other hand, we do not deeply study the reason why the kernel have three stages in the training process. Another limitation is that this paper does not investigate the general overfitting problem. We think both aspects will be important topics in our future works.

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

## A  PROOF

**Theorem 1** *Given a $\mathcal{C}^2$ function $f : \mathbb{R}^{n_{\text{in}}} \to \mathbb{R}^{n_{\text{out}}}$, parameterized by a vector of parameters $\theta$, and trained by $x' = x + \omega$, where $\omega \in \{-\epsilon, \epsilon\}^3$, on a dataset $\mathcal{D} = \{(x, y) | x \in \mathbb{R}^{n_{\text{in}}}, y \in \mathbb{R}^{n_{\text{out}}}\}$, NTK of $\mathcal{X}$ satisfies $\Theta(\mathcal{X} + \Omega_{t-1}, \mathcal{X})_{t-1} \approx \Theta(\mathcal{X}, \mathcal{X})_{t-1} + o(\epsilon)(\epsilon \to 0)$, where $\mathcal{X} = \{x | (x, y) \in \mathcal{D}\}$ and $\Omega_{t-1} = [\omega_1, \omega_2, \ldots, \omega_{|\mathcal{X}|}]$, which is a random perturbation set independent of $x$, $\theta$, and $\ell$.*

**Proof 1** *For a given $f$, we first calculate the changes in parameters $\theta$ and the outputs. We have*

$$\Delta(\theta)_t = \theta_t - \theta_{t-1} = -\eta \nabla_{\theta_{t-1}} f_{t-1}(\mathcal{X} + \Omega_{t-1})^T \nabla_{f_{t-1}(\mathcal{X} + \Omega_{t-1})} \mathcal{L}_{\text{AT}}$$
$$\Delta(f(\mathcal{X}))_t = f_t(\mathcal{X}) - f_{t-1}(\mathcal{X}) = \nabla_{\theta_{t-1}} f_{t-1}(\mathcal{X}) \Delta(\theta)_t,$$

*For $\Delta(f(\mathcal{X}))_t$, it can be rewritten as $-\eta \nabla_{\theta_{t-1}} f_{t-1}(\mathcal{X}) \nabla_{\theta_{t-1}} f_{t-1}(\mathcal{X} + \Omega_{t-1})^T \nabla_{f_{t-1}(\mathcal{X} + \Omega_{t-1})} \mathcal{L}_{\text{AT}}$.*

*We have*

$$\nabla_{\theta_{t-1}} f_{t-1}(\mathcal{X} + \Omega_{t-1}) \approx \nabla_{\theta_{t-1}} f_{t-1}(\mathcal{X}) + \Omega_{t-1}^T \nabla_{\mathcal{X}} \nabla_{\theta_{t-1}} f_{t-1}(\mathcal{X}).$$

*Without loss of generality, we suppose the over the $\Omega_{t-1}$, $\mathrm{P}(\omega^{i,j,k} = \epsilon) = p$. Then, we have*

$$\mathbb{E}(\omega^{i,j,k}) = \epsilon(2p - 1)$$
$$\mathbb{V}ar(\omega^{i,j,k}) = 4\epsilon^2 p(1 - p)$$

*Similarly, we suppose $g \in \nabla_{\mathcal{X}} \nabla_{\theta_{t-1}} f_{t-1}(\mathcal{X})$ obeys a distribution, whose expectation is $\mu_{t-1}$ and variance is $\sigma_{t-1}$. Usually, both $\mu_{t-1}$ and $\sigma_{t-1}$ are close to zero in a deep neural network.*

*Suppose $\omega^{i,j,k}$ and $g$ are independent, so $\mathbb{E}(\omega^{i,j,k}g) = \epsilon(2p-1)\mu_{t-1}$, $\mathbb{V}ar(\omega^{i,j,k}g) = 4\epsilon^2 p\sigma_{t-1}(1-\epsilon^2) + 4\epsilon^2 p(1-p)\mu_{t-1}^2 + 6\epsilon^2$. Based on Central Limit Theorem, the elements in $\Omega_{t-1}^T \nabla_{\mathcal{X}} \nabla_{\theta_{t-1}} f_{t-1}(\mathcal{X})$ approximately equal to the $\mathbb{E}(\omega^{i,j,k}g)$. Therefore, we have the following conclusion that, any element $\alpha$ in $\Omega_{t-1}^T \nabla_{\mathcal{X}} \nabla_{\theta_{t-1}} f_{t-1}(\mathcal{X})$ satisfies $\alpha = o(\epsilon)(\epsilon \to 0)$.*

*We have the conclusion that $\Theta(\mathcal{X} + \Omega_{t-1}, \mathcal{X})_{t-1} \approx \nabla_{\theta_{t-1}} f_{t-1}(\mathcal{X}) \nabla_{\theta_{t-1}} f_{t-1}(\mathcal{X})^T + o(\epsilon) = \Theta(\mathcal{X}, \mathcal{X})_{t-1} + o(\epsilon)(\epsilon \to 0)$.*

**Theorem 2** *Given a $C$-Lipschitz function $f : \mathbb{R} \to \mathbb{R}$ defined on a $l_2$-normed metric space, suppose $\mathcal{X} = \{x | x \in \mathbb{R}\}$ is the input set, whose cardinality is $|\mathcal{X}|$. Suppose there are $k$ disjoint subsets of $\mathcal{X}$ having the same size, i.e., $\tilde{\mathcal{X}}_i \in \mathcal{X}, i \in [k]$, and $|\tilde{\mathcal{X}}_i| = \alpha \leq |\mathcal{X}|$. Let $S_i^1 = \sum_{j=1}^{\alpha} x_j$ and $S_i^2 = \sum_{j=1}^{\alpha} x_j^2$, for $x_j \in \tilde{\mathcal{X}}_i$, and $A = S_i^2 - (S_i^1)^2$. For $x \in \tilde{\mathcal{X}}_i$, we have*

$$\|f(\frac{x - \tilde{\Sigma}_i}{\tilde{\Pi}_i}) - f(\frac{x - \Sigma}{\Pi})\|_2 \leq C\|x\|_2(\|\frac{1}{\tilde{\Pi}_i}\|_2 + \|\frac{\alpha}{\sqrt{\mathbb{E}(A)}}\|_2) + C(\|\frac{\tilde{\Sigma}_i}{\tilde{\Pi}_i}\|_2 + \|\frac{\mathbb{E}(S_i^1)}{\sqrt{\mathbb{E}(A)}}\|_2),$$

*where $\tilde{\Sigma}_i$ and $\tilde{\Pi}_i$ are mean and standard variance for elements in $\tilde{\mathcal{X}}_i$, and $\Sigma$ and $\Pi$ are unbiased estimators of mean and standard variance for elements in $\mathcal{X}$.*

**Proof 2** *As $f$ is a $C$-Lipschitz function, we have*

$$\left\|f(\frac{x - \tilde{\Sigma}_i}{\tilde{\Pi}_i}) - f(\frac{x - \Sigma}{\Pi})\right\|_2 \leq C\left\|\frac{x - \tilde{\Sigma}_i}{\tilde{\Pi}_i} - \frac{x - \Sigma}{\Pi}\right\|_2 = C\|x\|_2 \|\frac{1}{\tilde{\Pi}_i} - \frac{1}{\Pi}\|_2 + C\|\frac{\tilde{\Sigma}_i}{\tilde{\Pi}_i} - \frac{\Sigma}{\Pi}\|_2.$$

*We have the following relationships between $\tilde{\Sigma}_i$ and $\Sigma$ and between $\tilde{\Pi}_i$ and $\Pi$,*

$$\Sigma = \mathbb{E}(\tilde{\Sigma}_i)$$

$$\Pi = \sqrt{\frac{\alpha}{\alpha - 1} \mathbb{E}(\tilde{\Pi}_i^2)}.$$

*Thus, we have*

$$\left\|f(\frac{x - \tilde{\Sigma}_i}{\tilde{\Pi}_i}) - f(\frac{x - \Sigma}{\Pi})\right\|_2 \leq C\|x\|_2(\|\frac{1}{\tilde{\Pi}_i}\|_2 + \|\frac{1}{\sqrt{\frac{\alpha}{\alpha-1}\mathbb{E}(\tilde{\Pi}_i^2)}}\|_2) + C(\|\frac{\tilde{\Sigma}_i}{\tilde{\Pi}_i}\|_2 + \|\frac{\mathbb{E}(\tilde{\Sigma}_i)}{\sqrt{\frac{\alpha}{\alpha-1}\mathbb{E}(\tilde{\Pi}_i^2)}}\|_2)$$

$$\leq C\|x\|_2(\|\frac{1}{\tilde{\Pi}_i}\|_2 + \|\frac{1}{\sqrt{\mathbb{E}(\tilde{\Pi}_i^2)}}\|_2) + C(\|\frac{\tilde{\Sigma}_i}{\tilde{\Pi}_i}\|_2 + \|\frac{\mathbb{E}(\tilde{\Sigma}_i)}{\sqrt{\mathbb{E}(\tilde{\Pi}_i^2)}}\|_2)$$

$$= C\|x\|_2(\|\frac{1}{\tilde{\Pi}_i}\|_2 + \|\frac{1}{\sqrt{\mathbb{E}(\mathbb{E}(\tilde{\mathcal{X}}_i^2) - \tilde{\Sigma}_i^2)}}\|_2) + C(\|\frac{\tilde{\Sigma}_i}{\tilde{\Pi}_i}\|_2 + \|\frac{\mathbb{E}(\tilde{\Sigma}_i)}{\sqrt{\mathbb{E}(\mathbb{E}(\tilde{\mathcal{X}}_i^2) - \tilde{\Sigma}_i^2)}}\|_2)$$

$$= C\|x\|_2(\|\frac{1}{\tilde{\Pi}_i}\|_2 + \|\frac{\alpha}{\sqrt{\mathbb{E}(S_i^2 - (S_i^1)^2)}}\|_2) + C(\|\frac{\tilde{\Sigma}_i}{\tilde{\Pi}_i}\|_2 + \|\frac{\mathbb{E}(S_i^1)}{\sqrt{\mathbb{E}(S_i^2 - (S_i^1)^2)}}\|_2)$$

$$= C\|x\|_2(\|\frac{1}{\tilde{\Pi}_i}\|_2 + \|\frac{\alpha}{\sqrt{\mathbb{E}(A)}}\|_2) + C(\|\frac{\tilde{\Sigma}_i}{\tilde{\Pi}_i}\|_2 + \|\frac{\mathbb{E}(S_i^1)}{\sqrt{\mathbb{E}(A)}}\|_2)$$

## B EMPIRICAL STUDY OF MODEL INITIALIZATION

Through our study, the biggest difference between our work and previous works (Loo et al., 2022; Tsilivis & Kempe, 2022) is the assumption of the model initialization method. Specifically, in previous works, the common assumption of the neural network is that the parameters in the network obey a Gaussian distribution. It is aligned with some deep learning frameworks, such as JAX (Bradbury

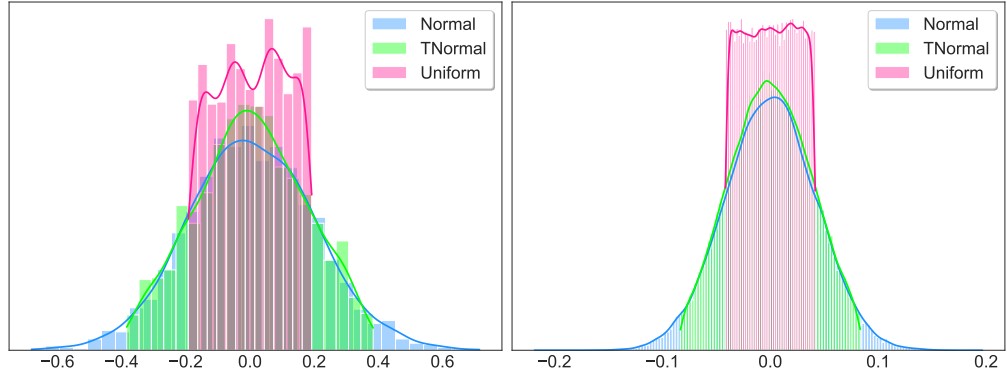

(a) Shallow Convolutional Layer Weight Distribution  (b) Deep Convolutional Layer Weight Distribution

Figure 11: Weight distributions under different initialization methods.

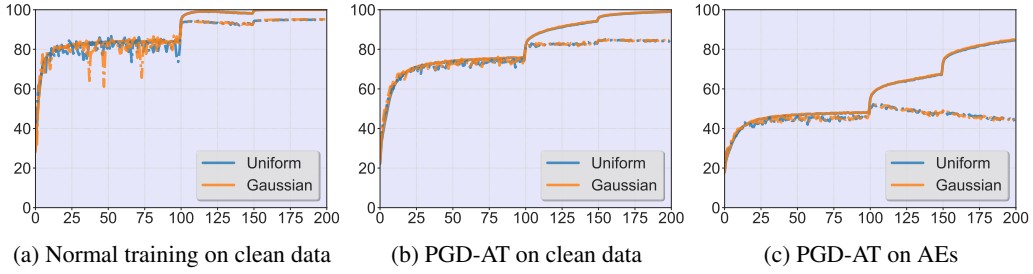

(a) Normal training on clean data  (b) PGD-AT on clean data  (c) PGD-AT on AEs

Figure 12: Accuracy under different initialization methods. The x-axis represents the training epoch. The y-axis represents the accuracy. Solid lines are for training set metrics. Dash lines are for test set metrics.

et al., 2018), in which parameters are initialized with a Gaussian distribution as a default setting. However, some deep learning frameworks, such as Pytorch (Paszke et al., 2019), adopt a Uniform distribution (He et al., 2015) to initialize all parameters as a default setting. As we do not assume any assumptions about the initialization method, it is important to explore the differences in kernel dynamics under different initialization methods.

In Figure 11, we plot the weight distributions of two convolutional layers under different initialization methods. There are three methods we consider, i.e., the Gaussian (Normal) Distribution, the Truncated Normal (TNormal) Distribution, and the Uniform Distribution. In Figure 11a, we show the weight distribution of a shallow 2D convolutional layer in ResNet-18, whose kernel size is 3, the input channel is 3, and the output channel is 64. In Figure 11b, we show the weight distribution of a deep 2D convolutional layer in ResNet-18, whose kernel size is 3, the input channel is 64, and the output channel is 128. The results show a lot of differences between different initialization methods. If these neural networks follow the "lazy training" pattern, they should be a linear approximate around their initial position. However, the following results empirically prove that finite wide networks do not really follow such a pattern.

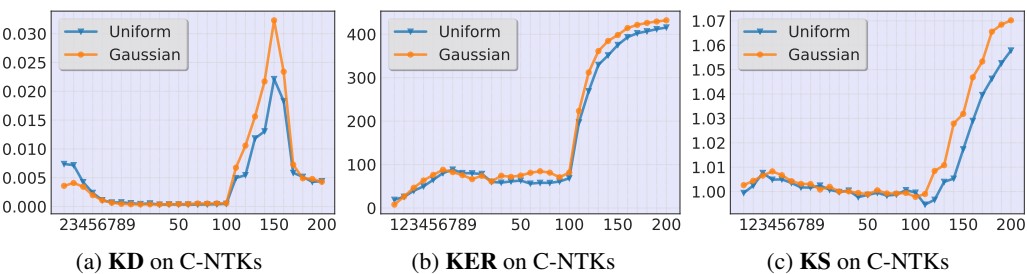

(a) **KD** on C-NTKs  (b) **KER** on C-NTKs  (c) **KS** on C-NTKs

Figure 13: Kernel dynamics of models trained under normal training.

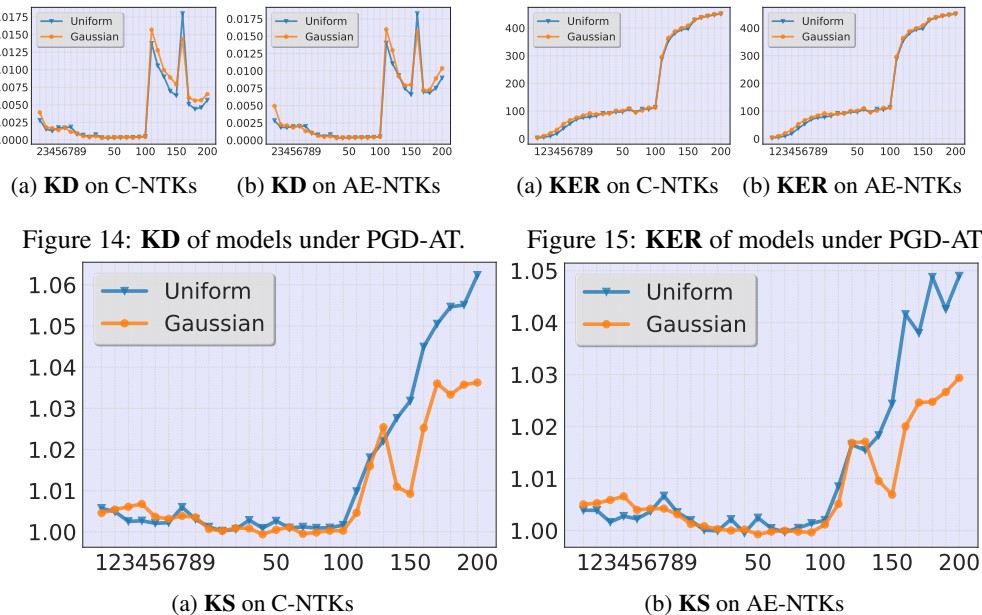

| (a) **KD** on C-NTKs | (b) **KD** on AE-NTKs | (a) **KER** on C-NTKs | (b) **KER** on AE-NTKs |

Figure 14: **KD** of models under PGD-AT.    Figure 15: **KER** of models under PGD-AT.

(a) **KS** on C-NTKs    (b) **KS** on AE-NTKs

Figure 16: **KS** of models trained under PGD-AT.

We follow the same experiment setting as in our main paper. The results in Figure 12 show that the performance of models under different initialization strategies is close to each other during the training process. On the other hand, in Figure 13, we compare the kernel dynamics for two initialization methods. The results indicate a similar kernel evolution process during the normal training in these two models. We further compare the kernel dynamics of models under PGD-AT in Figures 14, 15, and 16, and we have the same conclusion that the kernels have similar evolution process.

Since the kernel dynamics are similar and the model's performance is close to each other, a natural question is whether their kernels converge to the same one or the similar one. Surprisingly, the answer is negative. In Figure 17, we compare **KD** between kernels of different models, and we choose the Uniform initialization method as a baseline. As a result, the specialization strengths are different, which can be found in Figure 18. Therefore, it seems like that the kernel dynamics is the main reason that influences the model's performance, and the two models having different kernels can still have similar performance on their task.

Overall, our empirical studies of different model initialization methods prove that the parameter distribution only influences the learned kernel, instead of the kernel dynamics. And the kernel dynamics is the factor, which influence the model's performance on its task.

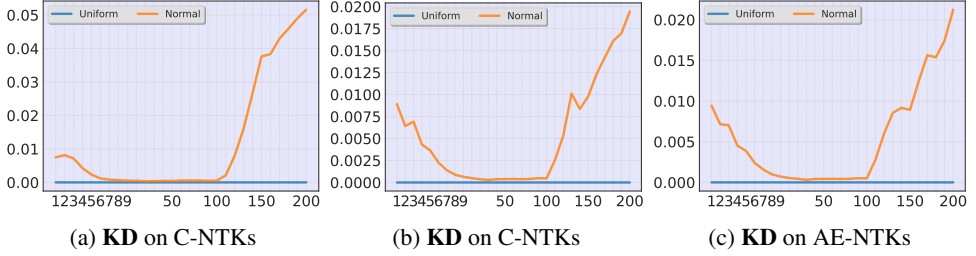

| (a) **KD** on C-NTKs | (b) **KD** on C-NTKs | (c) **KD** on AE-NTKs |

Figure 17: **KD** between different strategies.

## C    DISCUSSION OF REDUCING TRAINING COST

In Section 5, we show that the training time can be reduced by training the model with only clean data at an early stage. However, from the experiment results, we find that how to decide such a time point to introduce AEs into the training process is important. Clearly, there is a trade-off between

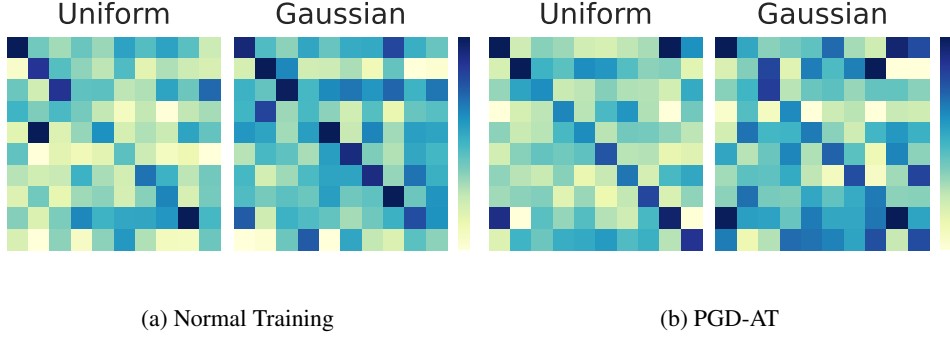

(a) Normal Training                                    (b) PGD-AT

Figure 18: $M_{\text{ks}}(\Theta_{200})$ on C-NTKs.

cost and robustness. If we adopt AEs to train the model at the end of the training process, usually we will not obtain a model with high robustness, although the training cost reduces a lot. On the other hand, if AEs are introduced too early, the training cost will stay high. Luckily, we find that under the training scheme proposed by Rice et al. (2020), the robustness can be obtained after the last learning rate decay if the number of training iterations is enough long. We evaluate models trained with different time reduce strategies under the PGD attack, C&W attack, and AutoAttack in Table 1.

Therefore, we recommend starting to use AT between the 110-th epoch and the 140-th epoch till the end of the training process. In this case, we can have a better trade-off between the training cost and robustness.

| Method | Time Cost (seconds) | Clean Accuracy | PGD-20 | PGD-100 | CW-100 | AA |
|--------|---------------------|----------------|--------|---------|--------|----|
| TE | 18000 | 81.93 | 55.17 | 54.82 | 52.50 | 50.50 |
| TE 90 | 12150 | 81.20 (0.33) | 54.66 (0.07) | 54.36 (0.01) | 51.51 (0.25) | 49.44 (0.08) |
| TE 100 | 11500 | 81.23 (0.20) | 54.16 (0.03) | 53.85 (0.06) | 51.04 (0.01) | 49.02 (0.11) |
| TE 110 | 10850 | 81.67 (0.06) | 54.24 (0.11) | 53.99 (0.12) | 51.14 (0.18) | 49.11 (0.11) |
| TE 140 | 8900 | 81.43 (0.25) | 52.82 (0.12) | 52.61 (0.11) | 49.43 (0.81) | 47.20 (0.57) |
| TE 150 | 8250 | 77.27 (0.45) | 49.37 (0.16) | 49.20 (0.22) | 45.81 (0.49) | 43.60 (0.54) |
| TE 160 | 7600 | 78.27 (0.22) | 49.60 (0.06) | 49.52 (0.06) | 45.20 (0.07) | 43.22 (0.12) |

Table 1: Evaluation results for different training strategies. We show the standard deviation in the brackets. The results indicate that our method to reduce training time costs is stable. Furthermore, even if we only spend about 10,000 seconds, the robustness and clean accuracy will only decrease a little, which is acceptable.

## D ADDITIONAL EXPERIMENTS

We consider a new training method, TRADES, in Figure 19. Compared to the other adversarial training strategies, TRADES shows very similar results on both C-NTKs and AE-NTKs under different metrics. The additional evaluation results prove our analysis provided in Section 3 is correct and general. The conclusions will not change for other adversarial training strategies.

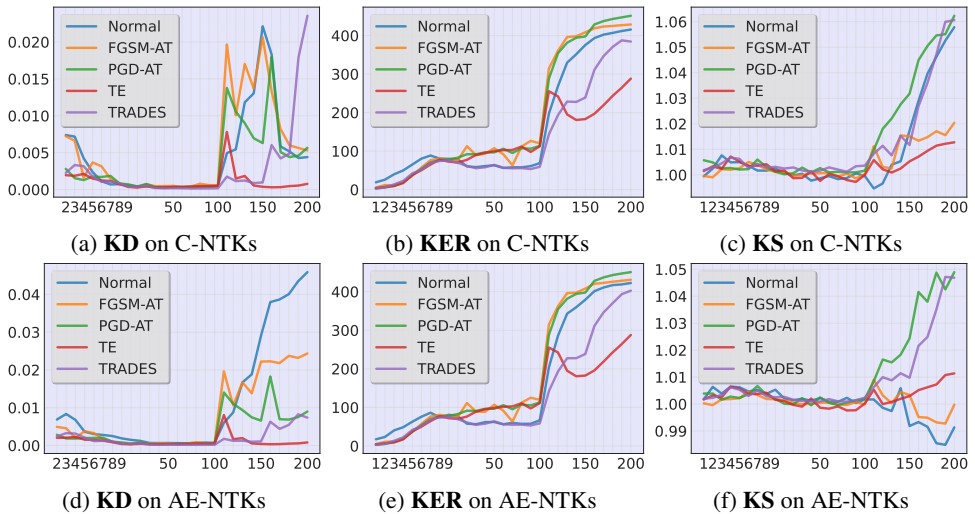

Figure 19: **KD**, **KER**, and **KS** in training. x-axis: training epoch; y-axis: metrics.

