# OpenReview forum: "Rethinking Adversarial Training with Neural Tangent Kernel"
_ICLR.cc/2024/Conference — ICLR 2024 Conference Withdrawn Submission_

### Official Review · Reviewer_rUEy · 2023-10-23

**Soundness:** 2 fair
**Presentation:** 2 fair
**Contribution:** 2 fair
**Rating:** 5
**Confidence:** 3

**Summary:**

This paper studies adversarial training from a kernel perspective, in which discussions about normalization, kernel dynamics, and spectral features are provided. The proposed method can reduce computation time, and experiments show the improvement compared to banchmarks.

**Strengths:**

The paper is clear and easy to understand.

**Weaknesses:**

[1] In the assumption at the beginning of Section 2.2, it imposes some assumption on the distribution of the attack. However, this is different from common understanding of the attack: given the model and the data, the attack should be the one to maximize the loss.

[2] In the proof of Theorem 1, in page 11, there is a "$\approx$" in one of the formulas. The authors need to exactly quantify what has been approximated in the formula, as some further constraints on $\epsilon$ may be required to ensure the correctness of this approximation. In particular, for common Taylor expansion approximations, L2 attack and L2 differences are commonly used. The authors need to justify the correctness of the approximation under Linf attack.

[3] Even if [2] is correct, it is still questionable whether the theoretical/technical contributions are novel enough: when $\epsilon$ is small enough, it is natural to use Taylor expansion to have approximation on the adversarial loss, so that one can treat the adversarial loss as a linear function of the adversarial attack to simplify the analysis. This assumption has been widely used in many theoretical literature in this area. From this perspective, the current presentation of Theorem 1 and Theorem 2 is not sufficient to demonstrate the technical challenges.

[4] Again for the assumption on $\epsilon$, I doubt whether $\epsilon=o(1)$ is a valid assumption or not. I agree that most theoretical literature considers $\epsilon=o(1)$ in their theory. However, when considering 8/255 Linf attack in CIFAR-10, the possible largest L2 norm of the attack is 8/255*sqrt(32^2) = 8*32/255 > 1, which means at least for L2 attack, a small attack strength is not sufficient to align with the real practice. In terms of the Linf attack in this paper, please fix [2] to see whether the total L2 strength can be $\Theta(1)$ or not.

[5] Missing comparison between the proposed method and other methods. The authors only compare their performance with the benchmark algorithm. However, there are some other algorithms which try to speed up the computation of adversarial training. For example, in

Jia, Xiaojun, et al. "Improving Fast Adversarial Training with Prior-Guided Knowledge." arXiv preprint arXiv:2304.00202 (2023).

Their proposed method is able to handle ImageNet dataset.

[6] There is a big gap between the theory and the experiments, and there is no clear connection on how to connect these two components in this paper.

**Questions:**

Please address my concerns in the Weakness section.

---

### Official Review · Reviewer_jZv6 · 2023-10-28

**Soundness:** 2 fair
**Presentation:** 1 poor
**Contribution:** 2 fair
**Rating:** 3
**Confidence:** 3

**Summary:**

The paper provides an error gap between the ground-truth NTK and empirical NTK in adversarial training, and observe NTK changes significantly in the AT process. The paper later provides three case studies.

**Strengths:**

The case study try to adopt Neural Tagent Kernel setting that proposed in Jacot et al 2018 into adversarial training setting, and provides some cases study under the NTK setting.

**Weaknesses:**

I didn’t find any connections between the theorem and the case studies. Theorem 1 is rather informal and the idea is straightforward. Theorem 2 lacks explanation and analysis on the upper bound. How should we see each of the terms in the upper bound, why ``the kernel of a model trained with mini-batch is symptotically closed to the kernel of a model trained directly on all data” is not clear to me. Does the author want to say the upper bound is small? Why? The upper bound is a fixed quantity that only depends on the property of the training samples. How does the bound actually relates to the result in the experiment?

In terms of the three metrics, they all introduced in previous work, therefore I cannot see any novelty from here. It’d be much more helpful if the author compared these three metrics from both the designed intuition (motivation) and the mathematical perspective. Also, can KD characterize the robustness of the model?

The dynamic of three-stage learning mainly comes from the learning rate decay. Therefore this might be an artifact of the current learning scheduler strategy, as if you choose a different one, for example, cyclic learning rate. The dymanic of the kernel metric might be different.

For case study 1, I don’t think it’s appropriate to use propositions 1 and 2 whereas they just remarks or observations, and the statement is not rigorous at all.

For case study 2, it seems standard training can be regarded as a warmup before adversarial training. However, how good the standard training depends on the number of iterations, which relates to the final robustness of the model, yet seems there’s no unified framework to set this parameter. Moreover, for different learning scheduler strategies, the proposed iteration might be different. I totally agree that it’s possible to reduce the training cost, but I don’t think the current content of case study 2 is enough to justify a complete story.

For case study 3, the author provides a simple solution without explaining the reason. It’s unclear to me why adding anisotropy noise can solve the problem, and does it only solve the problem in the AdvNTK setting or for the adversarial trained neural networks in general.

As the author mentioned in the limitation, the author does not dig into details regarding the three stages of the training procedure. I feel the author touches based on each problem slightly. The paper would be much stronger if one direction could be digged into deeply.

**Questions:**

Any particular reason why considering TE as the algorithm? I think the author should also give a brief introduction of what is TE and TE-OF. Such abbreviation without explanation is not explicitly clear.

I noticed in case study 1 the metric is KS, whereas case study 2 the metric changes to KD. Any reason how to choose different metric for different case studies?

---

### Official Review · Reviewer_9q81 · 2023-10-29

**Soundness:** 2 fair
**Presentation:** 1 poor
**Contribution:** 2 fair
**Rating:** 3
**Confidence:** 3

**Summary:**

The author investigated the properties and issues in Adversarial Training with the help of the Neural Tangent Kernel method. They focused on three aspects of adversarial training: the impact of using unbiased estimators in batch normalization layers on the model's robustness, clean examples to save training time inspired by kernel dynamics, and the cause of catastrophic overfitting problems from the kernel's perspective.

**Strengths:**

- Study an important problem of adversarial training and Neural Tangent Kernel.
- Provide extensive experiments across various AT methods, datasets, and metrics.

**Weaknesses:**

1. The paper's clarity requires improvement, as the overall writing can be challenging to follow. Consider segregating the presentation and discussion of theorems to enhance readability. Furthermore, the entanglement of background or related works in Section 2/3 with the proposed methods impedes a clear understanding of the paper's contributions. It would greatly benefit readers if a dedicated section were included for background and related works.

2. While the paper offers a comprehensive array of experiments, the complexity and lack of clarity in the proposed task make it difficult to discern the results. The alignment between the purpose of the experiments and the theoretical contributions is not clearly established, which raises concerns about the utility of the proposed theorem. Clarifying this linkage would enhance the paper's impact.

3. Would the author please further explain how to derive Proposition 1/2 from Theorem 2, as well as the proofs for these propositions?

4. In Section 5, the argument regarding the similarity of learned kernels from different training methods in the early stages and the use of clean examples for pre-training presents a reasonable alternative explanation for the pre-training process. However, the novelty of the proposed method may be limited, particularly given the common practice of adversarial training on pre-trained models. Moreover, the proposed kernel method can only decide when to start the AT empirically as indicated in Appendix C, which raises concern from a theoretical perspective. Could the threefold stage be used for deciding the time to AT?

5. In Figure 10 (b) FGSM-AT Modified, the robust accuracy drops to around 0 at epoch 100, which seems like an OC. But it recovers from the OC and it looks quite strange to me as OC usually can not self-recovery. Did the author run multiple experiments to avoid outliers or have any explanation for that?

**Questions:**

As stated in the Weakness.

---

### Official Review · Reviewer_2z5m · 2023-11-08

**Soundness:** 2 fair
**Presentation:** 2 fair
**Contribution:** 3 good
**Rating:** 3
**Confidence:** 3

**Summary:**

The paper studies the training dynamics of NN in the context of Adversarial Training. In particular, it focuses on the evolution of the Neural Tangent Kernel (NTK) and use it to get insight on various topics (batch normalization, training efficiency and catastrophic forgetting). It also proposes some improvements based on the NTK analysis.

**Strengths:**

* The problem discussed is overall interesting and the case studies make definitely a lot of sense (how to use insights from NTK in AT to improve some essential aspects of the robust learning process)
* I like the overall approach of using the NTK to establish some diagnosis and propose new approaches to improve the efficiency of Adversarial Training.
* Case Study III is promising ; It would be interesting to compare your approach with other ways to circumvent catastrophic forgetting
* Paper refers to main work on the field (Loo et al., 2022; Tsilivis & Kempe, 2022)

**Weaknesses:**

* [Presentation] The extensive usage of acronyms (sometimes very ad-hoc) makes the paper hard to read. Also Fig 6 and 10 are especially hard to read (maybe not all curves are needed in Fig 6 + it’s hard to distinguish dashed vs non dashed in some cases (even if we can guess which one is which based on the order of magnitude))
* Paragraph 2 (Error Gap Analysis) lacks clarity. It claims to study the error gap between the ENTK and the ground-truth NTK but rather focuses on relations betwen (clean sample) NTK and AE NTK. No discussion is provided and the relevance of Theorem 2 in this discussion is not clear to me.
* Motivations of the parameters in experiment configuration are unclear. Especially the scheduling of the learning rate at epochs 100/150. Also metric presentation could be improved (for instance, you might want to explain why KER was introduced and which type of behavior it is supposed to quantify)
* More discussion and results on the impact on the adversarial accuracy would be helpful to understand the impact of the strategies (especially some confidence interval on this quantity)

**Questions:**

* Could you give more detail regarding the "consequences" of Theorem 1/ 2 in Section 2.2 ? It is unclear to me how it helps to better understand the Error Gap ? Is the Theorem 2 only useful to back the discussion of Case III ?
* Can you give more context on the choice you made on the experiment setup ? Especially the choice of the learning rate scheduling. Is it standard to achieve a better abdversarial accuracy ?
* I would suggest the authors to improve some aspects of the presentation (acronyms, some figures not fully readable and some wordings (why 'w buffer' instead of 'w batch norm' ? am I missing an important distinction here ?))